# An Explorative Qualitative Study of the Role of a Genetic Counsellor to Parents Receiving a Diagnosis After a Positive Newborn Bloodspot Screening

**DOI:** 10.3390/ijns11020032

**Published:** 2025-04-28

**Authors:** Samantha A. Sandelowsky, Alison McEwen, Jacqui Russell, Kirsten Boggs, Rosie Junek, Carolyn Ellaway, Arthavan Selvanathan, Michelle A. Farrar, Kaustuv Bhattacharya

**Affiliations:** 1Graduate School of Health, University of Technology Sydney, Ultimo, NSW 2007, Australiaalison.mcewen@uts.edu.au (A.M.); 2Department of Clinical Genetics, Sydney Children’s Hospital, Randwick, NSW 2031, Australia; jacqui.russell@health.nsw.gov.au (J.R.); kirsten.boggs@mcri.edu.au (K.B.);; 3Genetic Metabolic Disorders Service, Syndey Children’s Hospital Network, Westmead, NSW 2145, Australia; 4Faculty of Medicine and Health, The University of Sydney, Westmead, NSW 2145, Australia; 5Department of Paediatric Neurology, Sydney Children’s Hospital, Randwick, NSW 2031, Australia; 6Discipline of Paediatrics and Child Health, School of Clinical Medicine, University of New South Wales Medicine and Health, UNSW, Randwick, NSW 2031, Australia

**Keywords:** newborn bloodspot screening, genetic counsellors, genetics, genomics, inborn errors of metabolism, spinal muscular atrophy, psychosocial adjustment, adaptation

## Abstract

Newborn Bloodspot Screening (NBS) can detect severe treatable health conditions with onset during infancy. The parents of a newborn baby are vulnerable in the days after birth, and the optimal way to deliver the shocking and distressing news of a potential serious diagnosis is yet to be defined. More data are needed to determine whether access to a genetic counsellor (GC) improves families’ experiences with genetic conditions identified by NBS. This study aimed to explore the similarities and differences for parents who received a positive NBS result for Spinal Muscular Atrophy (SMA) and received access to a GC (GC cohort), to a cohort of parents who received a diagnosis for inborn errors of metabolism (IEM) and did not have access to a GC (non-GC cohort). Semi-structured interviews explored the retrospective experiences of receiving the NBS result, including diagnosis implications and subsequent adaptation to respective genetic diagnoses. Inductive thematic analysis was used from group comparison. 7 SMA families and 5 IEM families were included in the study. Four themes were identified: 1. minimal pre-test counselling; 2. perceived lack of local healthcare team knowledge; 3. enabling factors for adaptation; 4. implications for both individuals and their families. Both the GC and non-GC cohorts reported insufficient counselling in the pre-test period and described feeling traumatised at the time of the diagnosis delivery. Families without subsequent GC input described limited understanding of the disease due to the use of medicalized terms, as well as a decreased understanding of reproductive options, familial communication and subsequent cascade screening. GCs can support information needs and adaptation following a NBS diagnosis.

## 1. Introduction

Newborn Bloodspot Screening (NBS) has been offered to babies in New South Wales (NSW) and the Australian Capital Territory (ACT), Australia, since the 1960s [1]. Currently, the test can detect more than fifty health conditions, the majority of which have a genetic basis [2]. In NSW, a heel-prick blood test is performed at 48–72 h of age, once parents have provided their consent with the relevant local healthcare team (LHCT), and most positive screen results are delivered by the 10th day of life. Clinical management after diagnostic confirmation, for most conditions, is implemented between day 10 and day 20 of life. Acute metabolic conditions such as maple syrup urine disease have interventions after the first screen test, with patients being managed before 10 days of life [3].

The principles underpinning the inclusion of conditions tested for by NBS are exemplified by the Wilson and Jungner criteria, developed on behalf of the World Health Organisation (WHO) [4]. In Australia, the NBS programme is an opt-in programme, estimated to screen over 99% of the total newborn population annually, numerically representing over 300,000 neonates. In NSW, screening commenced in the 1960s with PKU and subsequently congenital hypothyroidism. Cystic fibrosis and galactosaemia screening were introduced over the next twenty-five years [3,5,6]. The introduction of tandem mass spectrometry into public health screening in 1998 in NSW was a paradigm shift which dramatically increased the number of rare inborn errors of metabolism (IEM) that could be identified [7]. The implementation, centred around data for medium chain acyl co A dehydrogenase deficiency (MCADD), assumed that patients were identified earlier on in the disease course, hence clinical staffing did not change [8]. However, we can now determine that expanded NBS by TMS had metabolic perturbance that could be related to variants in over 50 different genes, and that genetic testing with appropriate counselling is increasingly relevant [1]. Furthermore, with much more genotypic and long-term prevalence data now available, some initially identified analytes may relate to a relatively benign conditions [9,10,11]. Similarly to genomic testing, these benign conditions and variants can be considered incidental findings that are not relevant to the original purpose of newborn screening [1].

Twenty years later, in 2018, SMA NBS was introduced after advocacy from a broad multi stakeholder group, including a well-established neuromuscular multiple disciplinary team (MDT), in NSW [12,13]. With novel therapeutics being introduced at the same time, SMA fulfilled the Wilson and Jungner criteria for an addition to NBS. The paediatric neuromuscular MDT incorporates specialist neurologists, nurses, physiotherapists, social workers and a genetic counsellor. For SMA, a specific workflow was developed to deliver timely urgent care [14]. The designated neuromuscular team is notified after the first positive SMA screen test. The team coordinates with LHCT to develop a family-centred plan for screen positive disclosure, including the delegation of roles for who is best placed to facilitate this process. The specialist and genetic counsellor support the information delivery at first notification. This new model of care differed from the established models for IEM, as before, the first screen test typically had a lower sensitivity and specificity [15,16,17]. For IEM after a positive screen test, a repeat heel-prick test and additional confirmatory tests are usually requested. Hence, for most IEM, such as PKU and MCADD, the screening laboratory would contact the LHCT to collect repeat tests, with some limited information about the target conditions and treatment being provided to the LHCT. The differences in standard NBS workflow between SMA and IEM are summarised in Figure 1.

The general implications of a positive NBS for a family may include both short and long-term psychological sequalae. Short term emotions identified after inconclusive cystic fibrosis screen results have included disbelief, shock, fear and uncertainty, leading to a sense of unpreparedness in managing their child’s health. Diagnostic delay with therapeutic uncertainty would trigger similar emotions for all screened conditions [18].

GCs are allied health professionals who are experienced in medical genomics, counselling and communication: they are well-placed to provide crisis counselling and support during the process of psychological adaptation to new diagnostic information for the family unit [19]. Furthermore, they are well-positioned within the multidisciplinary healthcare team to assist with the educational needs of the family regarding the condition.

The aim of this study was to explore the experience of two cohorts of parents who received a diagnosis of SMA with a GC or IEM without a GC after a positive NBS. SMA NBS was implemented in 2018 with the GC embedded as part of the MTD. Hence, the study was designed to test the difference in these models of care. Insights may be useful to establish the value of a GC, how they may be best embedded in a multidisciplinary team (MDT), and with further studies how the management pathway may be optimised to support families receiving a positive NBS result.

## 2. Materials and Methods

### 2.1. Study Design

This study was a qualitative cross-sectional study, using semi-structured interviews targeted at information delivery at the time of notification of the positive newborn screen test, as well as subsequent early diagnostic testing and follow up (Appendix A) [20]. Thematic analysis was performed on the qualitative data. Clinicians (consultant paediatric neurologists and metabolic specialists) identified families who received a positive NBS result for either SMA or an IEM for their first child, from 2018 to 2022. All children were under the age of five years, at the time of interview, to reduce recall bias after the positive screen test discussion and subsequent management. The SMA cohort had a genetic counsellor embedded in their service, whereas the IEM cohort had no genetic counselling, neither in the newborn period nor subsequent early childhood years. Table 1 summarises the IEM conditions of the families involved.

### 2.2. Recruitment

The selected families were contacted via email by a member of the research team. A follow-up email was sent two weeks later. Participants received an invitation, participant information sheet and a consent form (Appendix A). Families were able to partake when a signed consent form was returned via email, with a time for an online interview (30–45 min in duration) organised.

### 2.3. Data Collection

Online interviews were recorded via Microsoft Teams. A single researcher (SS) working under the supervision of senior research team members (K.B. (Kaustuv Bhattacharya), K.B. (Kirsten Boggs), J.R., A.M.) conducted the interviews. The interviews followed a guide developed from the clinicians’ expertise working in the newborn screening and paediatric acute care contexts (Appendix A). The interview guide included demographic questions.

### 2.4. Data Analysis

Due to the limited existing literature, an interpretivist epistemology was employed for this study [28]. The analysis approach was inductive, following reflexive thematic analysis methodology [29]. Transcripts were initially independently reviewed by investigators (SS and KBo), who collaboratively discussed the identified codes and emerging themes. The demographic information collected from the interview was analysed using descriptive statistics, including the mean for each parameter.

This study received ethics approval from Sydney Children’s Hospital Network (SCHN) Research Ethics Office (et 2022/ETH02031) and ratification from the University of Technology (ETH22-7765).

## 3. Results

A total of 28 invitations for interviews were sent (14 GC cohort and 14 non-GC cohort), with 12 interviews conducted (7 GC cohort and 5 non-GC cohort). Interviews were conducted with either one or two parents. Therefore, a total of 19 participants were involved in this study. The summary of the key demographics of the parents of the neonates are summarised in Table 2.

### 3.1. Themes

From the data, four themes were identified, which are summarised in Table 3.

#### 3.1.1. Theme 1: Minimal Pre-Test Counselling

The commonalities between both cohorts included a lack of understanding of the NBS, in particular the purpose of testing.


*So before [the heel prick test positive result] we didn’t know anything about the newborn screening or something like that, so when we left the hospital, they did the hearing check, sight check and checked all of that.*

*(GC cohort, Family 3, P5)*



*I wasn’t really aware what newborn screening was. They just approached me, we’re gonna take a prick of the foot. I’m not sure if it’s like mandatory in hospitals or not. So I just thought that’s just like a blood test.*

*(Non-GC cohort, Family 5, P7)*


Furthermore, parents reported that they were uncertain whether they had a choice for their infant not to have NBS.


*We are ‘what’s best people’ when it comes to the medical professionals and stuff like that, if they’re like well this is what we do then I’m just like, OK, well this is what we do.*

*(non-GC cohort, Family 1, P1)*



*Just went through the normal procedures that you do in hospital it’s just something that you do, right?*

*(GC cohort, Family 1, P2)*


When families received the NBS results they experienced denial and disbelief, particularly relating to the incongruency between the phenotype of their child and the information from the NBS.


*It’s like, how is the world still turning when we’ve got this information? And I’m just like, how is everyone acting like nothing has happened? But you just like my whole world’s crashing around me.*

*(GC cohort, Family 2, P3)*



*It was devastating. I think it was hard to believe it because he looked like a very normal, you know newborn. Physically there was no problem, it was a shock, I know I cried. It was hard. It was hard.*

*(GC cohort, Family 3, P6)*


The SMA cohort (GC cohort) described trepidation relating to the treatment options being foreign to them.


*It’s a big obviously decision to be making, putting your child’s life in, you know, in something that we don’t know about.*

*(GC cohort, Family 2, P3)*


Additionally, this cohort felt as though their pre-existing knowledge of biology and genetics was minimal, interfering with their ability to understand their child’s medical condition.


*Genetics was not something that I personally knew much about. Like I’d heard of cystic fibrosis before, but I didn’t know that that was a genetic condition.*

*(GC cohort, Family 7, P11)*


#### 3.1.2. Theme 2: LHCT Perceived Lack of Knowledge of Genetic Conditions

Both cohorts described the LHCT who were involved in the care of their infants to be uncomfortable and uncertain when dealing with a positive NBS result.


*Because [the LHCT] was uneducated about metabolic diseases, she was unable to even give me any information.*

*(non-GC cohort, Family 5, P7)*



*The GP we had at the time had no idea what SMA even was….yeah nobody had any idea of what we were talking about.*

*(GC cohort, Family 4, P7)*


#### 3.1.3. Theme 3: Factors Enabling Adaptation to Diagnosis

Both cohorts acknowledged psychosocial factors that impacted their response to a positive NBS result and subsequent genetic diagnosis. These included perinatal mental health, specifically post-partum depression, challenges with rural healthcare, previous adverse experiences with LHCTs and the educational backgrounds of parents.


*You know, I think like for me it’s the mental side. Yeah, it’s just like, yeah, it’s just mentally heavy.*

*(non-GC cohort, Family 2, P3)*



*More doctors like [our doctor, SHCT]. We didn’t have any doctors that knew what it is out in [rural NSW] even now. Our [LCHT] had to go away and look it up.*

*(non-GC cohort, Family 1, P1)*


Similarly, both cohorts described the benefit of specialist medical discussions at the time of their child’s diagnosis.


*Like I said, I’ve always we’ve always felt supported and nurtured and kind of protected.*

*(GC cohort, Family 5, P9)*



*They’re doing amazing. But just the information they’ve given us or they’re just being able to talk to someone. Couldn’t pick a better place to go to as I know I tell everyone that well, they have brilliant, but yeah, they looked after us.*

*(non-GC cohort, Family 2, P3)*


The non-GC cohort reported that there was a sense of confusion and uncertainty when they received information surrounding the genetic diagnosis, including accessibility of language and understanding the meaning of the condition.


*Like, I didn’t know what you were saying It’s all this [medical terminology]. Then he’s like this gene, this gene, like, they’re this diagram. Like, I don’t even know what you’re talking about.*

*(non-GC cohort, Family 4, P5)*


Contrastingly, the GC cohort described the emotional and information support they received from the healthcare teams and their support systems (family, friends and support groups). The GC cohort specifically referred to the value of having an MDT at diagnosis and made mention of the GC involved in their family’s care.


*Like [the genetic counsellor] being there, social worker being there, doctor being there (E: like it was good that, you felt as if it…)…probably the best part…because there’s a difference. Like everyone’s there, if you have any questions at the time, they could always be answered cause someone was always there to help answer it.*

*(GC cohort, Family 4, P7)*


#### 3.1.4. Theme 4: Implications from Diagnosis for Couples and Families

There was a key difference between the cohorts with respect to understanding the potential implication of the genetic condition identified through NBS. The GC cohort was able to understand the genetics of the condition, implications for family planning and options for familial cascade testing. This included the natural history of SMA including the relevant genotype, inheritance and treatments available to their child.


*It’s a degenerative disease. It can affect almost any part of your body, like your breathing, your swallowing, anything to do with your muscles, so practically everything.*

*(GC cohort, Family 4, P8)*



*If people [want to] know more of that explain about him missing the SMN1 gene. We can run off the backup of the SMN2 with the medication Spinraza, otherwise I explain [the option of] Zolgensma as well which replaces the SMN1 gene that he is missing essentially and is able to cross the blood brain barrier.*

*(GC cohort, Family 3, P6)*



*There is a one in four chance of being affected, so you got 25% transferring affected, 50% chance of being carrier and then 25% chance of being unaffected altogether, like neither of affected or a carrier of which [our daughter] actually is.*

*(GC cohort, Family 2, P4)*


From a psychosocial perspective, they described discussing their personal and familial guilt before sharing the news with relatives to undergo familial cascade screening.


*And we knew nothing about us both being carriers. I think for me, and that’s probably been the hardest part, is not knowing that we were a carrier of SMA. Obviously that was the most challenging thing.*

*(GC cohort, Family 7, P11)*


Furthermore, having a GC involved in their child’s care facilitated the consideration of value-based decision making for future pregnancies.


*So, we didn’t want to sort of test the theory, and then also you know I guess you come to a bit of a moral impasse as well, as to when like yes, you find the SMA like what do you do? Like [P3] would be very against terminating, so you know what’s the point really?*

*(SMA, P4) (GC cohort, Family 2, P3)*



*Like if we found out we were carrying a child with SMA, we would terminate that pregnancy. You know, that’s hard for me to say, but like, that’s the reality certainly going forward and you know, I regret nothing, about [my son], but that is just like what we would do.*

*(GC cohort, Family 5, P9)*


By contrast, understanding of the genetic risk appeared limited in the non-GC cohort.


*Would the condition stay the same in our next baby as it is with [our son]? Or would it worsen? Cause I think if it was to definitely be worse…… 100%, no way, we [would we] go through that again.*

*(non-GC cohort, Family 2, P4)*


## 4. Discussion

The aim of this study was to explore the impact of a GC on the experience of families who received a positive NBS result. Those with a diagnosis of SMA received GC support (GC cohort), whereas those with a metabolic disorder did not (non-GC cohort). Specifically, this study aimed to identify key educational and psychosocial needs for families who received positive NBS results. These have been identified in Table 4.

Both cohorts felt that they received minimal information prior to the NBS test and the potential outcomes. They perceived that information was limited to the technical aspects of the testing (i.e., blood collection from heel), but not on the likelihood of a positive diagnosis and further follow-up. Some families described the LHCT minimising potential outcomes from the NBS test. This sentiment is reflected in studies by Ulph et al. (2020) and Chudleigh et al. (2021), which highlighted that when parents feel uninformed, it can lead to long-term impacts on patient trust and engagement with the medical profession [30,31]. Some participants raised the point that communication during the prenatal period would be valuable, and resources such as being directed to specific pamphlets would improve their sense of confidence and understanding of the testing. However, there are resources available for NBS in NSW, but this study was not designed to assess recall of those resources.

Furthermore, parents found the purpose of the test to be elusive, hindering their ability to provide informed consent for their infant. This has been noted by Frankova et al. noting that parents understood the procedure rather than the purpose [32]. The notion of informed consent in this context highlights the idea of ‘inflicted ought’. This concept describes the experience for parents where they have the freedom and responsibility to choose a test for their infant but are hindered with the burden and guilt of that decision. Several families identified that if LHCTs such as midwives and nurses offered the test, they assumed that it was the best option for their child.

Parents in our study commented on their emotional distress when receiving a positive screening result, which they felt was exacerbated by the minimal counselling prior to the NBS. For 92% of families involved in the study, the communication with LHCTs occurred via phone call, which itself could hinder effective communication, particularly with such significant information. It is likely that members of the LHCT are unfamiliar with the condition being screened, knowing more detailed information would come with subsequent SCHT review [31]. This might be perceived by parents as evasiveness or non-empathetic, but it also reflects the paucity of outcome data in a range of rare conditions, meaning that uncertainty is often part of the communication process. However, there are opportunities for upskilling and further training in communicating with families of newborn babies [33]. From a genetics and genomics perspective, studies by Talwar et al. have emphasised upskilling medical and allied health professionals, which could provide much needed reassurance [34]. Families with false positive IEM diagnoses identified by TMS can exhibit significant distress for many months [35]. This reflects the fact that great care needs to be taken when a family is notified of a positive screen result, irrespective of what the final diagnostic outcome is.

Unlike many medical scenarios, a call for a positive newborn screening result, potentially indicating life-changing medical intervention is completely unexpected to carers of a seemingly well-looking baby. It is possible that there is no perfect way to make this call, as each family will have different factors determining resilience. The family have to adapt to clinical uncertainty, coming to terms with adverse implications [36]. Factors including baseline mental health, rural/remote healthcare, previous lived experience, and health literacy level have previously been identified to impact adaptation [37].

It should be noted that the critical difference in the model of care between the two cohorts studied was that the SCHT was involved supporting the disclosure of a positive screen for SMA, whereas the LHCT asked for confirmatory testing for IEM. The greater knowledge of SMA within the SHCT (supporting initial disclosure and ongoing care with the family) eliminated disorder-related uncertainty. By contrast for IEM, the LHCT was reported to have communicated the testing strategy with limited knowledge of the conditions they were testing. This is likely because there are approximately fifty conditions that could be identified by TMS screening and only SMA (albeit with varied phenotypes) can be identified by SMN1 copy number [17]. This patient cohort was also part of the early roll-out of SMA NBS. The SMA screening test has now been determined to have a sensitivity of 99%, meaning that specialist communication with the positive screen result was nearly always vindicated by the following confirmatory test discussion with the same team [38].

The SMA cohort described uncertainty relating to treatments, as there were minimal data on longer-term outcomes with early initiation of disease modifying therapies [38,39,40]. By contrast, NBS test for the IEM cohort did not necessarily confirm a specific diagnosis, leading to further biochemical and genetic testing with additional clinical follow-up. In addition to therapeutic uncertainty, there was diagnostic uncertainty with flow-on effects to the future for the child and family planning [41]. Genetic counsellors have specific training in genetic diagnostic uncertainty and could assist in helping families adapt during the investigation phase of screening.

As screening expands into the genomic era, with potential for an even greater number of genetic conditions to be identified, it will be essential that the LHCT, families, carers and community understand the process. This study demonstrates that the GC could play an important role in bridging the understanding and uncertainty that may follow.

The GC cohort highlighted the value of support groups and speaking with other families with the same genetic condition (via GC linkage), which improved social belonging and proved to be greatly effective for families who had minimal support from family and friends (Table 4). Plumridge et al. described the benefit of connecting families to the share their first-hand experiences of the condition [42]. This was a notable difference between the cohorts as the metabolic families identified support provided by the specialist medical team. The GC cohort could articulate the recurrence risk of SMA and explore their own individual beliefs and values to facilitate decision making surrounding family planning and prenatal testing. The literature has highlighted the importance of understanding the genetic conditions and couple-based decision making prior to communicating this information to other family members [43]. This was illustrated within the SMA cohort who received GC input when they communicated their carrier status for SMA to other at-risk relatives, to give them the option of cascade screening. Having these conversations with relatives often causes additional psychosocial consequences for parents including familial guilt and anxiety [44]. This greatly contrasted with the experience of the metabolic cohort (non-GC) who felt underequipped to communicate the outcomes to other family members because they had minimal understanding of basic genetic concepts. Furthermore, the group who had GC involvement had ongoing contact and support outside of the hospital setting, leading to better engagement with their entire healthcare team and regular reviews in outpatient departments. This was a clear distinction between the cohorts.

This study illustrates that access to a GC as part of the multidisciplinary specialist team does have a positive impact on families who receive a positive NBS result. GCs are well equipped to support the communication of complex medical and genetic information to families, in order to facilitate an understanding of the condition [45]. Depending on the context, some of the information can be delivered in a step-wise manner to assist in the comprehension of the condition, rather than information provision being communicated in a single event of breaking bad news, where misconception between the actual risk of a disease and diagnostic uncertainty may arise [45]. Furthermore, GCs assisted in the adaptation to the condition, which led to increased carer confidence [46]. As highlighted in this study, a GC can enable families to make value-based decisions for family planning and in communicating to other family members. From a psychosocial perspective, a GC can also signpost and refer families to support groups important for social acceptance [47]. A GCs involvement in the delivery of NBS results will function best as a part of an MDT SHCT, which has been summarised in Figure 2 [48,49,50].

The limitations of this study include the difference in sample sizes between the cohorts, meaning that there may not have been a broad representation of perspectives. The recruitment strategy utilised, limited the sample size. Future study designs may use alternative strategies such as hard copy invitations or opportunistic recruitment following clinical appointments [51]. Additionally, the IEM cohorts included a range of various conditions, which differed in terms of severity and management.

The differences in model of care could confound the outcome, as the timing of SCHT engagement was different. Furthermore, SMA screening was implemented with a MDT framework including a nurse, physiotherapist and social worker, as well as a GC, to manage high risk babies. The impact of the other members of the MDT has not been evaluated in this study. In contrast, the non-GC cohort with IEM has not evolved its MDT in line with contemporary genetic practice. The study attempted to eliminate recall bias by recruiting parents of children under five years of age. However, it is possible that some of the recurrence risk and genetic implications will have been covered in subsequent years by the GC in the SMA MDT. The IEM service had no embedded GC and none of the recruits saw a GC, neither in the newborn period nor subsequently. It is possible that the SMA cohort had a reinforcement of genetic knowledge in the early childhood years, whereas the IEM cohort did not. Nonetheless, the study highlights that adaptation to the clinical and genetic diagnosis would be enhanced with a GC as part of the MDT.

This study provides a sufficient base for future research to use quantitative methods (such as surveys) to explore the benefits of a GC when receiving NBS results. Longitudinal work is needed to track the progression of these families over time and to determine long term outcomes of those having access to a GC. Furthermore, with potential implementation of genomic NBS (gNBS), it is crucial that an effective genomic workforce team within rare disease centres of expertise is established, targeted to the needs of current families and for future families receiving positive genomic NBS results.

## 5. Conclusions

This study provides preliminary qualitative evidence that a GC involved in the care of families with NBS improves adaptation to the genetic diagnosis, as well as providing psychosocial support. Furthermore, the cohort who had a GC involved in their MDT felt confident with intra-familial communication for cascade testing. This research project provides options for further quantitative studies regarding family understanding and adjustment to the NBS diagnosis.

## Figures and Tables

**Figure 1 IJNS-11-00032-f001:**
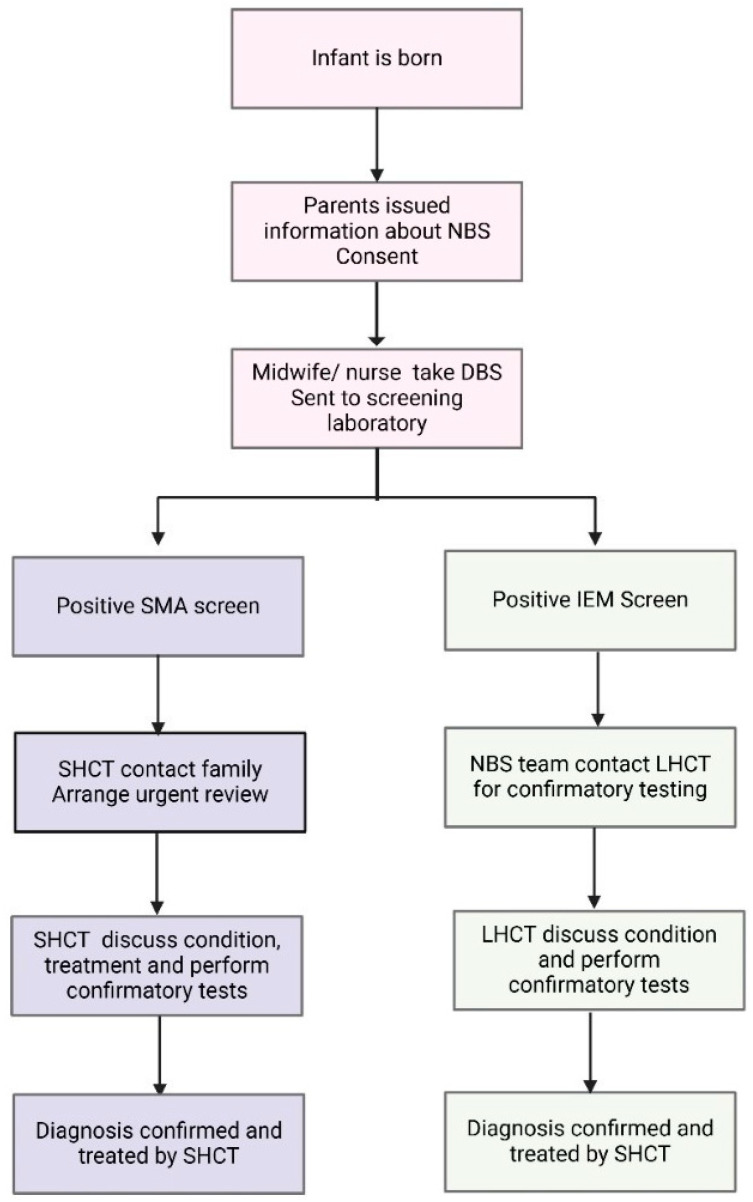
Schematic outlining the current NBS testing and management algorithm for SMA and IEM in NSW/ACT. The traditional model of care for IEM incorporates the local healthcare team (LHCT) in confirmatory testing, whilst the novel SMA algorithm has a specialist healthcare team (SHCT) involved in contacting the family and organising confirmatory testing.

**Figure 2 IJNS-11-00032-f002:**
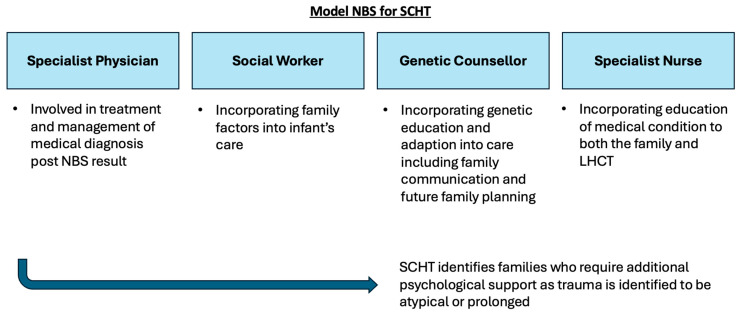
Summary schematic of proposed MDT model for SCHT for delivery of positive NBS results to families. The MDT includes a specialist physician, social worker, specialist nurse and genetic counsellor.

**Table 1 IJNS-11-00032-t001:** Summary of IEM conditions of cohort of families in this study.

Condition *	Phenotype	Treatment
Phenylketonuria (PKU)	Can lead to severe intellectual disability, seizures and behavioural problems [21].	Low phenylalanine diet; cofactor supplementation (Sapropterin). Normal outcome if treated [22].
Medium Chain Acyl-CoA Dehydrogenase deficiency (MCAD)	Hypoketotic hypoglycaemia with intercurrent illness, can be fatal [23].	Avoid prolonged fasting. carbohydrate-based sick-day plan. Normal outcome if treated [24].
Methylmalonic acidemia (MMA)	Acute metabolic decompensation (encephalopathy) with intercurrent illness. Developmental, cardiac and renal complications [25].	Low protein diet; carnitine supplementation, sick-day management; CNS, renal, cardiac and hepatic surveillance. Often progressive despite treatment [25].
Citrullinemia Type I	Acute metabolic decompensation (encephalopathy) [26].	Low protein diet, ammonia scavenger therapy, arginine supplementation, sick-day management—attenuated forms without symptoms may be identified [26].
3-methylgltaconyl-CoA hydratase deficiency	Ultra-rare leucine catabolic condition. Psychomotor delay, dystonia and metabolic acidosis reported but causality uncertain [27].	Dietary modification and sick day management uncertain [27].

* All IEMs are inherited in an autosomal recessive inheritance pattern.

**Table 2 IJNS-11-00032-t002:** Demographic characteristics of parent participants in the GC and non-GC cohort (n = 19).

	*GC Cohort (N = 12)*		Non GC Cohort (N = 7)	
*Sample Characteristics*	*n*	*%*	*n*	*%*
** *Gender* **				
*Male*	5	42	1	14
*Female*	7	58	6	86
18–24	1	8	-	-
25–31	1	8	-	-
32–38	8	67	6	86
39–45	2	17	1	14
** *Location of particpants* **				
Metropolitan	10	83	4	57
Rural/Remote	2	17	3	43
** *Age of child at time of interview* **				
0–12 months	2	17	4	57
12–24 months	3	25	3	43
2–3 years				
4–5 years	7	58		
** *Ethnicity of parents* **				
Aboriginal	1	8	1	14
European	7	58	6	86
African	2	17	-	-
Mediterranean	2	17	-	-
Middle Eastern				
** *Education level* **				
Completed School Yr 10/11	3	25	5	71
Finished High School (yr 12)	4	33	-	-
Undergraduate	4	33	2	29
Masters/PhD	1	8	-	-
** *Relationship Status* **				
Single				
Married	11	92	5	71
De facto	1	8	2	29

**Table 3 IJNS-11-00032-t003:** Summary of themes and subthemes identified in each study cohort.

Themes	Sub-Themes		
	Commonalities Between the Cohort	GC Cohort	Non-GC Cohort
Minimal pretest counselling.	Inflicted ought; lack of knowledge relating to NBS test; disbelief and denial of positive test result; lack of population knowledge about genetic condition.	Uncertainty surrounding treatment options.	Uncertainty surrounding the condition.
Perceived lack of LHTC education families who received positive NBS results.	Identified a lack of information and support and reassurance from LHCT who initially disclosed positive NBS outcome.		
Adaption to NBS diagnosis.	Various factors impacting receiving NBS diagnosis; Access to personal supports (family and friends).	Understanding of diagnosis, access to support groups, access to a GC.	Lack of understanding of NBS diagnosis due to medical terminology.
Implication for individuals and families.		Understanding of genetic condition, implications for family planning and communication with family members, value-based decision making.	Limited understanding of genetic condition.

**Table 4 IJNS-11-00032-t004:** Educational and psychosocial factors needed to navigate a genetic diagnosis.

*Factors*	
** *Educational* **	Understanding of the genetic condition including cause, symptoms, treatment and management, severity; Understanding inheritance and risk of recurrence; Understanding available reproductive options; Communicating to family members, option for familal cascade testing.
** *Psychosocial* **	Processing emotions associated with diagnosis/condition (including uncertainty, guilt, denial, grief and loss); Identifying social supports/social needs; Accessing support groups if desired (sense of belonging); Processing of cultural and religious considerations and factors relating to diagnosis.

## Data Availability

Confidential data is stored within secure server of Sydney Childrens’ Hospital’s Network. This is not publicly accessible but access can be requested from SCHN human research ethics committee.

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
