# Peer review of "An Explorative Qualitative Study of the Role of a Genetic Counsellor to Parents Receiving a Diagnosis After a Positive Newborn Bloodspot Screening"

_2409-515X, 2025, doi:10.3390/ijns11020032_

Round 1
Reviewer 1 Report
Comments and Suggestions for Authors
From the title, and the abstract I was hoping for a paper addressing the value of a GC in the first contact with the family after a NBS is reported as positive. But it becomes increasingly clear as I read the introduction and the methods and results that while there is an attempt to look back at the initial call, the vast majority of the paper is about the value of the GC in the care of the family with a diagnosis of a condition. The title is part of the problem “….a newborn bloodspot screening (NBS) Diagnosis”. NBS is screening. This is a paper about families who has a diagnosis of a conditions on the newborn screen after a positive screen. There is no such thing as “NBS diagnosis”.
The abstract states the issue well:
More data is needed to determine whether access to a genetic counsellor (GC) improves families’ experiences with genetic conditions identified by NBS. This study aimed to explore the similarities and differences for parents who received a positive NBS result for Spinal Muscular Atrophy (SMA) who received access to a GC (GC cohort), to a cohort of parents who received a diagnosis for an inborn errors of metabolism (IEM) and did not have access to a GC (non-GC cohort).
They are comparing experience of families who did and did not see a GC at the time of the initial NBS and diagnostic process, AND it becomes clear that this is not a discussion about just the NBS experience, since these are all families with an affected baby. Or at least that is my presumption, it is not actually stated. But assuming that is correct, they are comparing families’ understanding of (for example) “their reproductive options” on the basis of whether a GC was involved in the initial diagnostic process. Are we to presume that the New South Wales Genetics and Metabolism team never provided genetic counseling to their patients with PKU, MCADD, MMA, citrullinemia and 3-methylgltaconyl-CoA hydratase deficiency? This authors are telling us that having a GC involved leads to better understanding of genetics, and that makes sense, but it does not make sense that if a GC explained autosomal recessive inheritance to a family when the NBS is positive for PKU, before the diagnosis is made, that 3 years later they would understand their reproductive options…. How are the authors attributing the family’s knowledge at the time of the interview to the GC involvement?
Page 2 lines 54-60: this sentence is an important fact but extremely incompletely and awkwardly addressed. First of all what is meant by “several more children”? Typically “several” is similar to “a few”, like 3 or 4….. Does this meant several more a year? Several for each child who is going to be symptomatic? And in fact this is a statistical statement, and it should be clear that (for MCADD specifically which is what is here being discussed), it is not clear that we can say that this child needed to be identified and this did not. There are families in which one child with MCADD never had symptoms and another died as a toddler. This is a very messed up sentence, and I’m not sure that it has any relevance to the topic of the paper anyway, and unless it proves later that it is relevant, it should simply be removed. If this sentence needs to stay in as underpinning future discussion, it needs a lot of edits.
Page 2 line 67: “Because the first screening test …” ??? Same page above line 43 the authors describe that the sample for NBS “…is taken at 48-72 hours of age…” So what is the relevance of “first screen” here? Does this mean that the authors are going to tell us later that there is some repeat screening for IEM but not for SMA?
Page 2 line 73-80 AND the figure: I understand that the authors include experts in IEM. However, I will be extremely surprised if this is in fact the protocol for MCADD. MCADD is (at least in the USA) on the list of time-critical disorders, like MMA and MSUD. The list is posted on NewSTEPS at nEWBORN SCreening Quality Indicators In the United States, the parents of the baby with NBS positive for MCADD are instructed to avoid fasting/catabolism and diagnostic testing is done. We also typically go to diagnostic testing for PKU after a first screen, rather than a repeat screen, since the goal of treatment is to have treatment started by 2 weeks of life; if it takes 10 days to get a screen result then the baby will be nearly 3 weeks old by the time the second screen is done. I am not convinced this sentence and this table is an accurate description of the process and if it is an accurate description, then I think there may be some awkwardness in publishing it….
Page 3 line 89: the authors state they are citing a paper about the effect of NBS for CF on families (reference #19). However the paper they cite is one about the effect on parents of uncertainty about whether baby is affected after evaluation following NBS is inconclusive. Then in the next sentence, the authors state that prior studies of the effect of NBS on parents shows “increased potential for mental health problems, specifically anxiety and depression.” Interesting that the conclusion of the paper they cite actually says “Inconclusive CF NBS results were not associated with anxiety or vulnerability but led to health-related uncertainty and qualitative concerns.” If they really want to say that there is literature stating that parents have increased risk of depression after abnormal NBS, they should cite the literature. And, a fair citation of the literature would have to distinguish between literature about newborn screening itself (primarily literature around the effect of false-positive screen, with some concerns but mostly demonstration of resilience) and the effect of having a baby diagnosed with a significant condition after the NBS alerted to the diagnosis.
Page 6 in table 3, what is “inflicted ought”? (I see it is explained later)….
Page 6 in the table 3, I think there is going to be a problem with the identified theme; we are told that the SMA (with GC) cohort had uncertainty about the treatment options but the non-GC cohort had “uncertainty about the diagnosis”, and from this we are to presume that having a GC involved helped the GC cohort to have less uncertainty. BUT. The GC cohort got a call from the specialist team telling them there is a positive NBS and it is 99.9% sure baby is affected. There is little or no uncertainty about the diagnosis. The non-GC cohort (if we believe their description of the protocol) got a call from a primary health care provider to say there is a positive NBS and we don’t know if your baby had the condition and we need to do another NBS. So how would having a GC make that phone call resolve “uncertainty about the diagnosis”, when the initial contact with the family was for the purpose of telling them that there is uncertainty. If this is not a correct understanding of the process, then the manuscript needs to explain.
Page 9 the example of a failure of a non-GC cohort family to understand reproductive risks is given as: “Would the condition stay the same in our next baby as it is with [our son]? Or would it worsen? Cause I think if it was to definitely be worse…… 100%, no way, we [would we] go through that again. (non-GC cohort, Family 2, P4) Sorry, but that does not necessarily demonstrate lack of understanding. We would need context. This is not a family saying they don’t know the risk of recurrence; they may be very well aware it is 25%, and they could be giving an extremely cogent description of variable expression.
Page 9 lines 257-258: “The aim of this study was to explore the impact of a GC on the experience of families 257 who received a positive NBS result” If that is the aim then they have not identified the major source of bias, which is that these are all families interviewed after A DIAGNOSIS OF A CONDITION DISCOVERED BY NEWBORN SCREENING.
Page 10 line 280: The authors describe “inflicted ought” as an outcome, that the families they interviewed “… are hindered with the burden and guilt of” the decision to have NBs. This doesn’t make sense. “Inflicted ought” makes sense as a harm when one is studying false-positive NBS. But this this case, the family’s decision to have the NBS did lead them to a stressful situation, yes, but also to a diagnosis and the chance for treatment. I’m not sure there is much to talk about “inflicted ought” when the “inflicted ought” prevented intellectual disability or death….
Page 10 paragraph 283-297 is very nicely done but I don’t understand the second phrase in the sentence line 289-291: “ This might be perceived by parents as evasiveness or non-empathetic, but reflects uncertainty of specialist knowledge in a large range of rare conditions”. Is this saying that the primary care provider doesn’t have knowledge that the specialist has, or doesn’t know that the specialist has knowledge, or that the specialist is uncertain about what the specialist knows, or that the specialist knows there is uncertainty about the diagnosis or the prognosis? Otherwise very nice paragraph.
And the next paragraph is exactly right, and the job of the person making the call is to recognize the challenges and to provide the best possible support to each family.
Paragraph beginning line 314 this same page, finally the authors point out that the non-GC cohort not only didn’t talk to a GC at the time of the first call, they were specifically not told there is a diagnosis….. This needs to be clear from the beginning.
Page 11 paragraph beginning line 326, once again the authors are conflating the GC involvement in the NBS process and the GC involvement in the care of the family for the longer-term issues around understanding of risk of recurrence….
Page 11 line 346: “This study illustrates that access to a GC as part of the multidiciplinary specialist team does have a positive impact on families who receive a positive NBS screening result.” Certainly agree, though we do need to know that there is no GC involved in the metabolic team in order for us to say that this study illustrates the value of the GC in the team, we are only told there was no GC involved at the time of the newborn screening. And again, assuming there is no GC in the IEM team, this study does not demonstrate a positive impact on families who receive a positive NBS result, the authors (again assuming there is no GC associated with the metabolic team) demonstrate this difference between families with a child affected with a condition found by NBS who did and did not have opportunity to meet with a GC at the time they were notified about the positive screen and over subsequent months and years.
Page 12 the limitations of the study they cite the difference between cohort sizes (I’m not so concerned about that, 7 in one and 5 in the other) “and thus likely underpowered”. ?? What power? This is a descriptive study. I am no expert in statistics but I don’t think “power” is relevant.
Buried in the next paragraph, limitations include “In contrast the non-GC cohort with IEM has not evolved its MDT in line with contemporary genetic practice.” So it would appear that the authors are aware that there is a problem with the protocol after NBS.
But the authors do not include as a limitation the confounding effect of having years of follow up involving a GC and then reporting that outcomes as the effect on the GC on the NBS process.
The authors conclude:
This study provides preliminary qualitative evidence that a GC involved in the care of families with NBS improves adaptation to the genetic diagnosis, as well as providing psychosocial support. Furthermore, the cohort who had a GC involved in their MDT felt confident with intra-familial communication for cascade testing. This research project provides options for further quantitative studies regarding family understanding and adjustment to the NBS diagnosis
I think the authors need to be clear they are providing evidence that GC involvement in care of families not “with NBS” but “with conditions diagnosed as a result of NBS” is helpful. Overall this could be a terrific addition to the literature, but there needs to be clarity about what is a result of the GC involvement in immediate follow up of NBS (the initial communication of the positive screen and the initial diagnostic process) and what is the result of the GC involvement in the subsequent care; this will involve more clarity around the usual clinical practices, and the confounding effects, and addressing the specific concerns above.
Comments on the Quality of English Language
Some significant and annoying problems with grammar (or typos?) throughout; for example page 2 line 45 is missing a verb, line 71 has a comma after “specialist” that needs to be removed…
Author Response
Thank you for the thoughtful review and the nature of it clearly demonstrates that we have to much more to distinguish the models of care between the two departments, and also provide more detail on the conduct of the study. To that end, we have included as appendix one, the semi-structured guide for interview indicating that the questions were focused about the newborn period, when putative diagnoses were being established. We have avoided going into too much detail about the conditions, but like all metabolic conditions, there is huge variability in any given condition so a particular diagnosis (such as citrullinemia – in this case very mild) may not necessarily follow the trajectory published in the literature. So a diagnosis does not necessarily predict a phenotype. We now know there is a stronger genotype-phenotype alignment with SMA, but that fact only became apparent over the time course of these children’s lives. All families interviewed had children identified with a condition by NBS, confirmed on subsequent testing. The children were all under the age of 5 years (to minimise recall bias).
Metabolic patients are managed differently to USA by metabolic paediatricians and not necessarily embedded within genetic units (as they are in USA). Australia’s model of care for metabolic patients is more like European models than North American models of care. None of the metabolic patients had any access to a genetic counsellor at birth or to the age that these interviews occurred. SMA NBS was implemented in 2018 with the GC embedded as part of the MTD. Hence the study was designed to test the difference in these models of care.
From the title, and the abstract I was hoping for a paper addressing the value of a GC in the first contact with the family after a NBS is reported as positive. But it becomes increasingly clear as I read the introduction and the methods and results that while there is an attempt to look back at the initial call, the vast majority of the paper is about the value of the GC in the care of the family with a diagnosis of a condition. The title is part of the problem “….a newborn bloodspot screening (NBS) Diagnosis”. NBS is screening. This is a paper about families who has a diagnosis of a conditions on the newborn screen after a positive screen. There is no such thing as “NBS diagnosis”.
Appendix 1 indicates the structure of the interview and the focus was to assess the impact of the GC in the first communications rather than ongoing care.
The title has been changed to:
An explorative qualitative study of the role of a genetic counsellor to parents receiving a diagnosis after a positive newborn bloodspot screen. (NBS).
The abstract states the issue well: More data is needed to determine whether access to a genetic counsellor (GC) improves families’ experiences with genetic conditions identified by NBS. This study aimed to explore the similarities and differences for parents who received a positive NBS result for Spinal Muscular Atrophy (SMA) who received access to a GC (GC cohort), to a cohort of parents who received a diagnosis for an inborn errors of metabolism (IEM) and did not have access to a GC (non-GC cohort). They are comparing experience of families who did and did not see a GC at the time of the initial NBS and diagnostic process, AND it becomes clear that this is not a discussion about just the NBS experience, since these are all families with an affected baby. Or at least that is my presumption, it is not actually stated. But assuming that is correct, they are comparing families’ understanding of (for example) “their reproductive options” on the basis of whether a GC was involved in the initial diagnostic process. Are we to presume that the New South Wales Genetics and Metabolism team never provided genetic counseling to their patients with PKU, MCADD, MMA, citrullinemia and 3- methylgltaconyl-CoA hydratase deficiency? This authors are telling us that having a GC involved leads to better understanding of genetics, and that makes sense, but it does not make sense that if a GC explained autosomal recessive inheritance to a family when the NBS is positive for PKU, before the diagnosis is made, that 3 years later they would understand their reproductive options…. How are the authors attributing the family’s knowledge at the time of the interview to the GC involvement?
We have tried to limit recall bias by asking parents of pre-school children ie under 5 years of age and focused our questions around the newborn period per appendix one.
I have also added this statement to methods.
“All children were under the age of five years, at time of interview, to reduce recall bias after the positive screen test discussion and subsequent management. The SMA cohort had a genetic counsellor embedded in their service whereas the IEM cohort had no genetic counselling, neither in the newborn period nor subsequent early childhood years.”
Page 2 lines 54-60: this sentence is an important fact but extremely incompletely and awkwardly addressed. First of all what is meant by “several more children”? Typically “several” is similar to “a few”, like 3 or 4….. Does this meant several more a year? Several for each child who is going to be symptomatic? And in fact this is a statistical statement, and it should be clear that (for MCADD specifically which is what is here being discussed), it is not clear that we can say that this child needed to be identified and this did not. There are families in which one child with MCADD never had symptoms and another died as a toddler. This is a very messed up sentence, and I’m not sure that it has any relevance to the topic of the paper anyway, and unless it proves later that it is relevant, it should simply be removed. If this sentence needs to stay in as underpinning future discussion, it needs a lot of edits.
Determining benign and severe variants and gene discussion is entirely relevant to the potential role of a genetic counsellor. I have rephrased this: ” However ,we can now determine that expanded NBS by TMS had metabolic perturbance that could be related to variants in over 50 different genes and that genetic testing with appropriate counselling is much more relevant in the diagnostic phase. Furthermore, with much more genotypic and long-term prevalence data now available, some initially identified analytes may relate to a relatively benign condition. Similar to genomic testing, these benign conditions and variants can be considered incidental findings, that are not relevant to the original purpose of newborn screening.”
Page 2 line 67: “Because the first screening test …” ??? Same page above line 43 the authors describe that the sample for NBS “…is taken at 48-72 hours of age…” So what is the relevance of “first screen” here? Does this mean that the authors are going to tell us later that there is some repeat screening for IEM but not for SMA?
The reviewer is correct and this has been removed.
Page 2 line 73-80 AND the figure: I understand that the authors include experts in IEM. However, I will be extremely surprised if this is in fact the protocol for MCADD. MCADD is (at least in the USA) on the list of time-critical disorders, like MMA and MSUD.
I agree and this is what is said line 79- 81,” For severe acute IEM such as maple syrup urine disease, the IEM specialist is typically involved after the first sample is reported, in order to institute urgent treatment.” MCADD comment below
The list is posted on NewSTEPS at nEWBORN Screening Quality Indicators In the United States, the parents of the baby with NBS positive for MCADD are instructed to avoid fasting/catabolism and diagnostic testing is done. We also typically go to diagnostic testing for PKU after a first screen, rather than a repeat screen, since the goal of treatment is to have treatment started by 2 weeks of life; if it takes 10 days to get a screen result then the baby will be nearly 3 weeks old by the time the second screen is done. I am not convinced this sentence and this table is an accurate description of the process and if it is an accurate description, then I think there may be some awkwardness in publishing it….
The senior author is an experienced metabolic specialist, and the protocol is summarised in line 73 -80. The screening system is different in USA compared to many other countries and indeed PKU KPI’s for intervention are by day 20 rather than day 14 – in other countries such as UK the time for intervention in PKU is even later. There is no evidence from literature that any of this difference in timings has a material difference in clinical outcome. For all fat oxidation patients (MCADD, LCHAD, VLCADD, MTP, CPT1a and CPT2), the repeat test notification goes out with instructions to local medical team to feed the baby four hourly and to contact the local medical team if there is an acute sickness. The local medical team are instructed to contact the on-call metabolic team if there is an emergency. This protocol has served NSW well for 27 years, with one MCAD death prior to screen result on day 2 (unlikely to have been identified by any screening programme.) Part of the problem in Australia, is that there are only 5 screening laboratories in an area just smaller than USA. The metabolic services are co-located with the 5 screening labs and confirmatory testing facility in a public health model. There are no other metabolic services. It’s very difficult to relocate screen positive individuals across thousands of miles so we have different systems to USA, where there are many more state-based laboratories and more metabolic services. I did not think was relevant to the paper. However, I am happy to include this if requested.
Page 3 line 89: the authors state they are citing a paper about the effect of NBS for CF on families (reference #19). However the paper they cite is one about the effect on parents of uncertainty about whether baby is affected after evaluation following NBS is inconclusive. Then in the next sentence, the authors state that prior studies of the effect of NBS on parents shows “increased potential for mental health problems, specifically anxiety and depression.” Interesting that the conclusion of the paper they cite actually says “Inconclusive CF NBS results were not associated with anxiety or vulnerability but led to health-related uncertainty and qualitative concerns.” If they really want to say that there is literature stating that parents have increased risk of depression after abnormal NBS, they should cite the literature. And, a fair citation of the literature would have to distinguish between literature about newborn screening itself (primarily literature around the effect of false positive screen, with some concerns but mostly demonstration of resilience) and the effect of having a baby diagnosed with a significant condition after the NBS alerted to the diagnosis.
The point is taken – the reference was kept in because some of the metabolic diagnoses are uncertain diagnoses – such as mild Citrullinaemia and 3-methylglutaconic hydratase deficiency ie the natural history of the biochemical finding is unknown so we thought the relevance to uncertain CF diagnosis data was relevant and have clarified the sentence further to reflect this.
“Short term emotions identified after inconclusive cystic fibrosis screen results have included disbelief, shock, fear and uncertainty, leading to a sense of unpreparedness to manage their child’s health. Diagnostic delay with therapeutic uncertainty would trigger similar emotions for all screened conditions.”
We have removed the next part of the introduction and have used a different citation to discuss this further in the discussion.
Page 6 in table 3, what is “inflicted ought”? (I see it is explained later)….
Page 6 in the table 3, I think there is going to be a problem with the identified theme; we are told that the SMA (with GC) cohort had uncertainty about the treatment options but the non-GC cohort had “uncertainty about the diagnosis”, and from this we are to presume that having a GC involved helped the GC cohort to have less uncertainty. BUT. The GC cohort got a call from the specialist team telling them there is a positive NBS and it is 99.9% sure baby is affected. There is little or no uncertainty about the diagnosis. The non-GC cohort (if we believe their description of the protocol) got a call from a primary health care provider to say there is a positive NBS and we don’t know if your baby had the condition and we need to do another NBS. So how would having a GC make that phone call resolve “uncertainty about the diagnosis”, when the initial contact with the family was for the purpose of telling them that there is uncertainty. If this is not a correct understanding of the process, then the manuscript needs to explain.
SMA screening was only implemented in 2018 in some centres worldwide – NSW was one of the earliest adopters. We can now say retrospectively that we are 99% sure that low copy number is associated with severe disease, but did not know that prospectively for this cohort born between 2018 and 2023. The real difference, for the purposes of this paper, is the model of care in delivery of the information. The support the GC provided (as well as the SCHT) has helped to work through this phase of early adoption to a novel screening process. Prospectively we did not know the certainty of the screening test for SMA but used a structured communication system to support communication.
Because of the potential confusion in the difference between retrospective sensitivity determination and prospective knowledge in the implementation phase of SMA screening from 2018, I have reserved commentary of 99% sensitivity for SMA only to discussion section and not in introduction.
Page 9 the example of a failure of a non-GC cohort family to understand reproductive risks is given as: “Would the condition stay the same in our next baby as it is with [our son]? Or would it worsen? Cause I think if it was to definitely be worse…… 100%, no way, we [would we] go through that again. (non-GC cohort, Family 2, P4) Sorry, but that does not necessarily demonstrate lack of understanding. We would need context. This is not a family saying they don’t know the risk of recurrence; they may be very well aware it is 25%, and they could be giving an extremely cogent description of variable expression.
For the conditions described in this paper and the way the information would have been conveyed is that there is genotype-phenotype correlation. The difference in communication was that the information would have been delivered by a metabolic physician in the non-GC cohort and a GC in the GC cohort. Most often, conditions with variable expression or incomplete penetrance are not intended to be screened in the newborn period. The main exception to that is ABCD1 (ALD) screening which has not yet been implemented in Australia.
Page 9 lines 257-258: “The aim of this study was to explore the impact of a GC on the experience of families 257 who received a positive NBS result” If that is the aim then they have not identified the major source of bias, which is that these are all families interviewed after A DIAGNOSIS OF A CONDITION DISCOVERED BY NEWBORN SCREENING.
Sorry that is loose language and we have tightened that up
“The aim of this study was to explore the impact of a GC on the experience of families who received a diagnosis after a positive newborn bloodspot screen.” The title of the paper has also changed to clarify this commentary:
An explorative qualitative study of the role of a genetic counsellor to parents receiving a diagnosis after a positive newborn bloodspot screen (NBS).
Page 10 line 280: The authors describe “inflicted ought” as an outcome, that the families they interviewed “… are hindered with the burden and guilt of” the decision to have NBs. This doesn’tmake sense. “Inflicted ought” makes sense as a harm when one is studying false-positive NBS. But this this case, the family’s decision to have the NBS did lead them to a stressful situation, yes, but also to a diagnosis and the chance for treatment. I’m not sure there is much to talk about “inflicted ought” when the “inflicted ought” prevented intellectual disability or death….
The inflicted ought refers to consent for NBS and the parental memory of the consent process independent of the outcome of the test. The sentiment was common even though some of the metabolic cohort had no risk of intellectual disability or death.
Page 10 paragraph 283-297 is very nicely done but I don’t understand the second phrase in the sentence line 289-291: “ This might be perceived by parents as evasiveness or non empathetic, but reflects uncertainty of specialist knowledge in a large range of rare conditions”. Is this saying that the primary care provider doesn’t have knowledge that the specialist has, or doesn’t know that the specialist has knowledge, or that the specialist is uncertain about what the specialist knows, or that the specialist knows there is uncertainty about the diagnosis or the prognosis? Otherwise very nice paragraph. And the next paragraph is exactly right, and the job of the person making the call is to recognize the challenges and to provide the best possible support to each family.
Paragraph beginning line 314 this same page, finally the authors point out that the non-GC cohort not only didn’t talk to a GC at the time of the first call, they were specifically not told there is a diagnosis….. This needs to be clear from the beginning.
That context of specialist knowledge really refers to all of the scenarios mentioned, reflecting lack of data on very rare scenarios such as 3-methylglutaconic hydratase deficiency – we have had only one case after screening 2.5 million babies over 25 years. I cannot specifically mention this as the family can be re-identified but have rephrased this as:
“This might be perceived by parents as evasiveness or non-empathetic, but reflects paucity of outcome data in a range of rare conditions, meaning that uncertainty has to be part of the communication process.”
Page 11 paragraph beginning line 326, once again the authors are conflating the GC involvement in the NBS process and the GC involvement in the care of the family for the longer-term issues around understanding of risk of recurrence….
The first GC the metabolic cohort talked to, was the student conducting this study. This is clarified in methods. Of note, I found this quote in Susan Weisbren’s TMS screening paper from USA in 2003: “Genetic counselors, rarely consulted, also may provide valuable reproductive counseling and information.” Clearly that has had an impact in USA but not Australia.
Page 11 line 346: “This study illustrates that access to a GC as part of the multidiciplinary specialist team does have a positive impact on families who receive a positive NBS screening result.” Certainly agree, though we do need to know that there is no GC involved in the metabolic team in order for us to say that this study illustrates the value of the GC in the team, we are only told there was no GC involved at the time of the newborn screening. And again, assuming there is no GC in the IEM team, this study does not demonstrate a positive impact on families who receive a positive NBS result, the authors (again assuming there is no GC associated with the metabolic team) demonstrate this difference between families with a child affected with a condition found by NBS who did and did not have opportunity to meet with a GC at the time they were notified about the positive screen and over subsequent months and years.
As above – we do cite the Weisbren paper in this discussion and have tried to minimise bias by focusing the discussion around the first few weeks of life and interviewing in the first 5 years of life.
Page 12 the limitations of the study they cite the difference between cohort sizes (I’m not so concerned about that, 7 in one and 5 in the other) “and thus likely underpowered”. ?? What power? This is a descriptive study. I am no expert in statistics but I don’t think “power” is relevant.
Buried in the next paragraph, limitations include “In contrast the non-GC cohort with IEM has not evolved its MDT in line with contemporary genetic practice.” So it would appear that the authors are aware that there is a problem with the protocol after NBS. But the authors do not include as a limitation the confounding effect of having years of follow up involving a GC and then reporting that outcomes as the effect on the GC on the NBS process.
As above – hopefully appendix 1 will assist in determining where we tried to aim discussion. Yes you are right – power ids over stated.
The authors conclude: This study provides preliminary qualitative evidence that a GC involved in the care of families with NBS improves adaptation to the genetic diagnosis, as well as providing psychosocial support. Furthermore, the cohort who had a GC involved in their MDT felt confident with intra-familial communication for cascade testing. This research project provides options for further quantitative studies regarding family understanding and adjustment to the NBS diagnosis
I think the authors need to be clear they are providing evidence that GC involvement in care of families not “with NBS” but “with conditions diagnosed as a result of NBS” is helpful. Overall this could be a terrific addition to the literature, but there needs to be clarity about what is a result of the GC involvement in immediate follow up of NBS (the initial communication of the positive screen and the initial diagnostic process) and what is the result of the GC involvement in the subsequent care; this will involve more clarity around the usual clinical practices, and the confounding effects, and addressing the specific concerns above.
We have altered discussion to say this
“The study attempted to eliminate recall bias by recruiting parents of children under five years of age. However, it is possible that some of the recurrence risk and genetic implications will have been covered in subsequent years by the GC in the SMA MDT. The IEM service had no embedded GC and none of the recruits saw a GC, neither in the newborn period nor subsequently. It’s possible that the SMA cohort had reinforcement of genetic knowledge in the early childhood years whereas the IEM cohort did not.”
This leads naturally on to the concluding sentence. Nonetheless, the study highlights that adaptation to the clinical and genetic diagnosis would be enhanced with a GC as part of the MDT.
Comments on the Quality of English Language Some significant and annoying problems with grammar (or typos?) throughout; for example page 2 line 45 is missing a verb, line 71 has a comma after “specialist” that needs to be removed…
Reviewer 2 Report
Comments and Suggestions for Authors
This is a well written and interesting paper to read, thank you.
However, there is an important difference between a positive NBS result and a diagnostic outcome that needs to be made explicit throughout - even in the title. This is sometimes blurred but they are really important differences. Even though the specificity of the SMA NBS is explained well, as noted, confirmatory testing is still required as the SMN1 copy number is so important in terms of potential disease severity etc.,
It is helpful that the differences are made clear in Figure 1 for SMA and IEM in terms of confirmatory testing. However, confirmatory testing is still required. Therefore, I wonder if the comparison is between when GCs are introduced at the time of a positive NBS result as opposed to once confirmatory testing has been undertaken and a definitive diagnosis has been made.
I appreciate these difference seem subtle but they are extremely important especially when communicating outcomes with families. Therefore, I think it is imperative that it is expressed correctly throughout the paper.

Author Response
However, there is an important difference between a positive NBS result and a diagnostic outcome that needs to be made explicit throughout - even in the title.
Title is changed to An explorative qualitative study of the role of a genetic counsellor to parents receiving a diagnosis after a positive newborn bloodspot screen.
This is sometimes blurred but they are really important differences. Even though the specificity of the SMA NBS is explained well, as noted, confirmatory testing is still required as the SMN1 copy number is so important in terms of potential disease severity etc.
Per reviewer 1 comments, the discussion around specificity is moved to discussion to avoid confusion about the role-out of SMA screening, but it does help determine the difference between disease related uncertainty in IEM versus SMA.
It is helpful that the differences are made clear in Figure 1 for SMA and IEM in terms of confirmatory testing. However, confirmatory testing is still required. Therefore, I wonder if the comparison is between when GCs are introduced at the time of a positive NBS result as opposed to once confirmatory testing has been undertaken and a definitive diagnosis has been made.
We have added in appendix one to provide further information around what we were trying to ascertain. From retrospective recall, it is difficult to know how much the family's determined the difference between screen positive and diagnosis positive testing. However, support of the whole process appears to have made a difference - I have changed language around this.
I appreciate these difference seem subtle but they are extremely important especially when communicating outcomes with families. Therefore, I think it is imperative that it is expressed correctly throughout the paper.
Completely understand that we need to be explicit in our communication and appreciate the advice.
Round 2
Reviewer 1 Report
Comments and Suggestions for Authors
This reviewer very much appreciates the improvements in the presentation of this interesting study; all my concerns were adequately addressed and I look forward to seeing it published.